# Unusual polarimetric properties for interstellar comet 2I/Borisov

S. Bagnulo 1,11 ✉, A. Cellino 2,11, L. Kolokolova3,11, R. Nežič 1,4,5, T. Santana-Ros 6,7, G. Borisov 1,8, A. A. Christou1, Ph. Bendjoya9 & M. Devogèle10

So far, only two interstellar objects have been observed within our Solar System. While the first one, 1I/'Oumuamua, had asteroidal characteristics, the second one, 2I/Borisov, showed clear evidence of cometary activity. We performed polarimetric observations of comet 2I/Borisov using the European Southern Observatory Very Large Telescope to derive the physical characteristics of its coma dust particles. Here we show that the polarization of 2I/Borisov is higher than what is typically measured for Solar System comets. This feature distinguishes 2I/Borisov from dynamically evolved objects such as Jupiter-family and all short- and long-period comets in our Solar System. The only object with similar polarimetric properties as 2I/Borisov is comet C/1995 O1 (Hale-Bopp), an object that is believed to have approached the Sun only once before its apparition in 1997. Unlike Hale-Bopp and many other comets, though, comet 2I/Borisov shows a polarimetrically homogeneous coma, suggesting that it is an even more pristine object.

[1] Armagh Observatory & Planetarium, College Hill, Armagh, UK. [2] INAF – Osservatorio Astrofisico di Torino, Pino Torinese, Italy. [3] Department of Astronomy, University of Maryland, College Park, MD, US. [4] Mullard Space Science Laboratory, Department of Space & Climate Physics, University College London, Dorking, Surrey, UK. [5] Centre for Planetary Sciences, University College London/Birkbeck, London, UK. [6] Departamento de Fisica, Ingeniería de Sistemas y Teoría de la Señal, Universidad de Alicante, Alicante, Spain. [7] Institut de Ciències del Cosmos (ICCUB), Universitat de Barcelona (IEEC-UB), Barcelona, Spain. [8] Institute of Astronomy and National Astronomical Observatory, Bulgarian Academy of Sciences, Sofia, Bulgaria. [9] Université Côte d'Azur, Observatoire de la Côte d'Azur, CNRS, Laboratoire Lagrange, Nice, France. [10] Arecibo Observatory, University of Central Florida, Arecibo, PR, USA. [11] These authors contributed equally: S. Bagnulo, A. Cellino, L. Kolokolova. ✉email: stefano.bagnulo@armagh.ac.uk

Comet 2I/Borisov, also known as C/2019 Q4 (Borisov), was discovered on 30 August 2019 by Gennady Borisov at the MARGO observatory, Crimea (MPEC 2019-R106). Its orbital eccentricity (3.356191 ± 0.000015) shows that the object is not gravitationally bound to the Solar System, making it the first unambiguous case of a comet arriving from interstellar space. This is only the second recognised case of a small body entering the Solar System from interstellar space. The first such case, namely that of 1I/'Oumuamua, discovered on 18 October 2017 by the Pan-STARRS 1 telescope, was that of an apparently asteroidal body, lacking detectable signs of cometary activity[1]. In contrast, 2I/Borisov exhibited a coma, and its apparition provided a unique opportunity to glean information about a body apparently similar to Solar System comets, but otherwise unrelated to them. Most observations to-date were aimed at obtaining spectra of the comet, and at measuring its dust and gas emissions, in order to determine the material composition and compare it with those of Solar System comets. In particular, reflectance spectra in the 0.49–0.92-µm wavelength range show a reddish slope[2] similar to that of other Solar System bodies, including comets, Jupiter Trojan asteroids belonging to the so-called D taxonomic class[3] and several Centaurs and transneptunian objects[4] (the latter two categories exhibiting a wide range of colours, including – but not limited to – bodies with extremely red spectral slope).

Measurements of cometary linear polarisation provides information about physical characteristics of the coma dust particles that is difficult to obtain by other observing techniques. Sunlight scattered by dust particles is partially polarised, i.e., the electric field associated to the radiation has a preferred plane of oscillation. The polarised fraction of the radiation varies with the scattering angle, and depends on the characteristics of the scattering medium, in particular its complex refractive index (hence its chemical composition), and morphology, that includes size distribution, shape and structure of the scattering particles. In planetary science, polarisation is measured as the flux perpendicular to the plane Sun-Object-Observer (the scattering plane) minus the flux parallel to that plane, divided by the sum of the two fluxes; this measurement is usually repeated in different viewing conditions described by the so-called phase angle (the angle between the directions to the Sun and to the observer as seen from the target). Surprisingly, surfaces of airless objects (like the Moon or asteroids) and cometary atmospheres show similar phase angle dependence of polarisation. In particular it is found that at small phase angles (≤20°), the linear polarisation is directed along the scattering plane; because the way the polarisation is measured, this situation is described as negative polarisation. At larger phase angles, the linear polarisation becomes positive, that is, directed along the direction perpendicular to the scattering plane, and increases until it reaches its maximum at phase angles ≃90–100°. The devil is in the details, and the difference between different kinds of Solar System bodies, as well as between specific comets or asteroids, can be seen from values and location of minimum and maximum polarisation, from the phase angle values where negative polarisation changes to the positive one, or from the wavelength gradient of the polarisation[5–8].

Extraction of the characteristics of the scattering medium from the polarimetric results is a difficult task as it requires modelling of numerous light-scattering phenomena such as reflection, diffraction, interference, shadow hiding, etc. Specifically, the negative polarisation observed for airless bodies results mainly from so-called coherent backscattering, an effect resulted from multiple scattering between the regolith particles[9,10], whereas in cometary atmospheres single scattering and, thus, the properties of the dust particles themselves, define the observed polarisation. The composition, size and structure of the cometary particles vary not only from comet to comet, but also within each comet; the particles are different near the nucleus and in the tail, in jets and ambient coma, close to the Sun and far from it[8,11]. Intensive numerical computation[12,13] and laboratory data[14–16] are used for the modelling of the observations. A remarkable result of polarimetric investigations was the prediction that cometary dust is made of aggregates of submicron grains[17], which was later confirmed by the in situ studies of the Rosetta mission[18] and of the samples returned by the Stardust mission[19].

Even without numerical modelling, some information may be readily inferred through a very simple analysis of the physics of light scattering. For instance, light scattered by a low-albedo surface or complex (aggregated) particles tends to be more polarised than the light scattered by a higher-albedo surfaces or aggregated particles (the so-called Umov effect)[20], as high-albedo scatterers are more likely to produce multiple scattering, which in turn is responsible for a more efficient depolarisation. Larger positive polarisation can be also associated with smaller, and, thus, Rayleigh-like, particles, while light scattered by more complex large aggregated particles is more affected by multiple scattering and, thus, is more depolarised. In case of comets, the continuum polarisation may be strongly modified by molecular emission lines due to the gas component within the coma and tail, therefore broadband polarimetric measurements should be analysed taking also into account the spectral characteristics of the object, or even better, using filters specially designed to avoid emission lines.

Accurate polarisation-phase curves have been obtained for a number of comets and over a wide range of phase angle, from 0° up to more than 100° in some cases, depending on target distance from the Sun and from the Earth in different circumstances[8]. It is therefore of the highest interest to understand whether a comet coming from interstellar space shares the same kind of polarimetric behaviour exhibited by Solar System comets, since, in principle, comets that accreted in other astrophysical environments could be significantly different from Solar System bodies.

Here we report polarimetric observations obtained with the European Southern Observatory (ESO) Very Large Telescope (VLT), and we show that the polarisation of interstellar comet 2I/Borisov is quite different to what is generally observed in comets of our Solar System, with a notable exception, that of comet C/1995 O1 (Hale-Bopp). Hale-Bopp is believed to have appeared close to our Sun only once before its recent approach in 1997, therefore its material is quite pristine, but its polarimetrically homogeneous coma suggests an even more pristine nature for comet 2I/Borisov.

## Results

Data were obtained in service mode using the FORS2 instrument[21] attached at the Cassegrain focus of the Unit 1 (Antu) of the ESO VLT. FORS2 was used in imaging polarimetric mode with the V_HIGH filter (centred at 557 nm with a 123-nm full-width half-maximum, or FWHM), R_SPECIAL (centred at 655 nm with a 165-nm FWHM) and I_BESS (centred at 768 nm with a 138-nm FWHM). Reflectance spectra of comet 2I/Borisov show no prominent emission lines in the range covered by the R_SPECIAL and I_BESS filters[2,22] (hereafter referred to as $R_F$ and $I_F$ filters, respectively), but the V_HIGH filter (hereafter referred to as $V_F$ filter) covers exactly the second brightest emission after the CN(0-0) at 387 nm – the $C_2$ Swan band at 512 nm. However, comet 2I/Borisov appears to be $C_2$-depleted[23], so the $V_F$ filter is also not severely contaminated by molecular emissions. Therefore we assume that the observed broadband polarisation is due to the dust properties.

| DATE | UT | $t_{exp}$ | FILTER | $r$ | $\Delta$ | $\alpha$ | $P_r = Q/I$ | $U/I$ | Notes |
|---|---|---|---|---|---|---|---|---|---|
| | | (s) | | (a.u.) | (a.u.) | (°) | (%) | (%) | |
| 2019-12-25 | 08:02 | 880 | $V_F$ | 2.04 | 1.94 | 28.48 | 3.35 ± 0.07 | −0.01 ± 0.07 | – |
| 2019-12-25 | 08:24 | 480 | $R_F$ | – | – | – | 3.76 ± 0.08 | 0.17 ± 0.08 | – |
| 2019-12-25 | 08:39 | 360 | $I_F$ | – | – | – | 4.14 ± 0.14 | −0.03 ± 0.14 | – |
| 2020-01-08 | 08:05 | 1280 | $V_F$ | 2.12 | 1.95 | 27.56 | 2.59 ± 0.06 | 0.14 ± 0.06 | – |
| 2020-01-08 | 08:34 | 720 | $R_F$ | – | – | – | 3.32 ± 0.18 | 0.20 ± 0.10 | a |
| 2020-02-06 | 08:17 | 1280 | $I_F$ | 2.40 | 2.12 | 24.14 | 1.37 ± 0.28 | −0.88 ± 0.32 | b |
| 2020-02-06 | 08:46 | 1440 | $R_F$ | – | – | – | 1.19 ± 0.21 | 0.03 ± 0.21 | b |
| 2020-02-17 | 04:12 | 1280 | $I_F$ | 2.54 | 2.21 | 22.63 | 1.14 ± 0.18 | 0.11 ± 0.18 | – |
| 2020-02-17 | 04:24 | 1440 | $R_F$ | – | – | – | 0.53 ± 0.19 | −0.08 ± 0.14 | – |
| 2020-03-20 | 03:14 | 1120 | $V_F$ | 3.02 | 2.53 | 18.06 | −1.05 ± 0.15 | 0.09 ± 0.16 | – |
| 2020-03-20 | 03:41 | 1440 | $R_F$ | – | – | – | −0.82 ± 0.13 | 0.25 ± 0.10 | – |

**Table 1 Observing log and polarimetric measurements of comet 2I/Borisov.**

Table is organised as follows: observing date and time (cols. 1 and 2); exposure time (col. 3); the filter used for the observations (col. 4); the heliocentric distance $r$ and geocentric distance $\Delta$ to the comet (cols. 5 and 6, respectively); the phase angle (col. 7); the reduced Stokes parameters $Q/I$ and $U/I$ (cols. 8 and 9). Column 10 refers to special table footnotes. Effective wavelength and bandwidths as follow: filter $V_F$ is centred at 557 nm with a 123-nm FWHM; filter $R_F$ is centred at 655 nm with a 165-nm FWHM; filter $I_F$ is centred at 768 nm with a 138-nm FWHM.
aA background star was very close to the comet photocentre during the exposures with retarder waveplate at PA = 22.5°, 45°, 67.5°, which were discarded from the analysis.
bData were obtained with very poor seeing and in a very crowded background. See notes in the text.

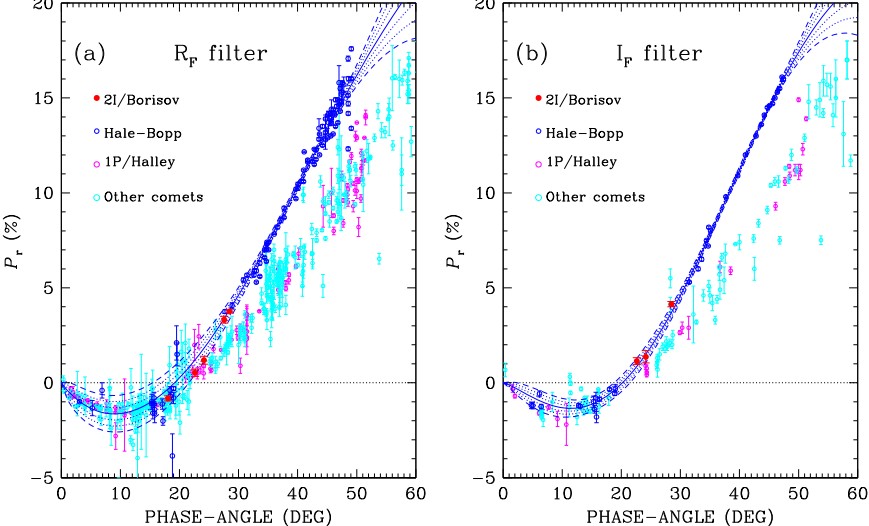

**Fig. 1 Broadband polarimetry of 2I/Borisov and other comets.** Panel **a** refers to the $R_F$ filter, and panel **b** to the $I_F$ filter. Red filled circles: data for 2I/Borisov. Empty blue, magenta and light blue circles: data for C/1995 O1 (Hale-Bopp)[11,38,73–76], comet 1P/Halley[26] and other comets[26], respectively. All data points are plotted with 1$\sigma$ error bars, calculated as explained in Methods subsection Polarimetry. The blue solid lines represent the best-fit model to the Hale-Bopp data obtained with Eq. (1); dotted lines show the ±1$\sigma$ and ±2$\sigma$ uncertainties, dashed lines show its ±3$\sigma$ uncertainties.

The reduced $Q/I$ and $U/I$ Stokes parameters[24] were obtained using aperture polarimetry and rotated to the reference direction perpendicular to the scattering plane (see Methods subsection Polarimetry for details). In this system of reference, $Q/I$ represents the flux perpendicular to the Sun-Comet-Earth plane minus the flux parallel to that plane, divided by the sum of the two fluxes (that is, the total flux). For symmetry reasons, the reduced Stokes parameter $U/I$ should be close to zero. In the following, instead of $Q/I$, we will adopt the often-used notation $P_r$.

Our data covers a fairly large interval of phase angles, including the critical transition around the inversion angle, but does not sample the negative branch: additional observations in the negative branch were scheduled at the VLT in April and May 2020, but due to COVID-19, science operations on Paranal were suspended. In addition to the aperture polarimetry described above, we also employed a slit polarimetry method[25] to measure polarisation along the comet tails, and we obtained also photometric measurements of the coma.

**Aperture polarimetry.** The results of our aperture polarimetry measurements are given in the observing log of Table 1, and data obtained in the $R_F$ and $I_F$ filters are shown in Fig. 1, together with the polarisation measurements of comet Hale-Bopp, of comet 1P/Halley, and of other comets, selected with the help of an online database[26]. Polarimetric observations of comets are obtained in filters with different effective wavelength and bandwidth, and using different apertures. In Fig. 1 we show literature data obtained with filters with different bandwidths, all centred at wavelengths between 620 and 695 nm for comparison with data taken with the $R_F$ filter (panel a), and between 715 and 810 nm for comparison with data taken with the $I_F$ filter (panel b). Some photometric filters employed by the observers may be designed to cover certain gas emission bands, or to cover only the dust continuum, or may cover both regions potentially dominated by molecular lines and regions dominated by the dust continuum, like those that we have employed for our observations. For our comparison with literature we have avoided using data obtained

through filters designed to cover molecular bands, yet the presence of molecular bands may well affect measurements with broadband filters. In an attempt to minimise these effects, when comet polarisation obtained at a given epoch was reported for different apertures, we adopted the value corresponding to the smallest one, on the grounds that it should be the most representative of the inner coma dust, where usually the highest values of polarisation are observed[8]. Some outliers with very large error bars have also been omitted.

Our observations show that the polarisation-phase curve of 2I/Borisov demonstrates a regular behaviour, with a clearly monotonic and linear increase of polarisation for increasing phase angle. Like comets of our Solar System, also comet 2I/Borisov exhibits negative polarisation at small phase angles. The value of the inversion angle helps to distinguish some classes of Solar System objects characterised by unusual properties. For instance, inversion angle values as large as 27–30° characterise a class of spinel-rich asteroids named Barbarians after the prototype (234) Barbara[27]. On the lower end of the observed range are the asteroids belonging to the F-taxonomic class[28,29], that have an inversion angle $\alpha_{inv}$ between 15° and 17°, a property shared also by the nucleus of comet 2P/Encke observed in the absence of coma[30] and by active asteroid 133P/Elst-Pizarro[31]. Centaurs seem to have more peculiar phase-polarisation curves, with small $\alpha_{inv}$ values, in particular centaur Chiron shows $\alpha_{inv}$ between 6.5° and 8° (ref. [32]). Active comets instead exhibit a homogeneous behaviour, with $\alpha_{inv}$ generally around 22° (ref. [8]). Comet 2I/Borisov is not an exception, with $\alpha_{inv} \simeq 20.5°$ (in the $R_F$ filter), a value slightly smaller than for other comets, but still consistent with the average in our Solar System. The slope at the inversion angle is another important characteristic of the polarimetric curve, and it is empirically found that for asteroids there exists a relationship between the slope of the polarimetric curve at the inversion phase angle and the albedo – the steeper the slope the lower the albedo[6]. From a linear interpolation of the measurements of 2I/Borisov in the $R_F$ filter we obtain a slope of $0.45 \pm 0.03$ % deg$^{-1}$. This value is somewhat extreme for comets, that exhibit polarimetric slopes typically ranging between 0.2 and 0.4 % deg$^{-1}$[8]. If our target were an asteroid, a slope of 0.45% deg$^{-1}$ around the inversion angle would correspond to an extremely low albedo (<0.05). However, the albedo-polarisation relationship cannot be strictly applied to cometary comae and tails, because of the very different physics of light scattering by rather densely packed layers of particles in the former, and by rarified clouds of particles in active comets.

Figure 1 suggests a strong similarity between the polarisation behaviour of 2I/Borisov and that of comet C/1995 O1 (Hale-Bopp), that may be better illustrated by performing a fit to the data. Polarimetric curves of the small bodies of the Solar System may be fitted with the empirical function[33,34]

$$P_r(\alpha) = b \, (\sin \alpha)^{c1} \left( \cos \frac{\alpha}{2} \right)^{c2} \sin(\alpha - \alpha_0) \qquad (1)$$

where $\alpha$ is the phase angle, and $b$, $c1$, $c2$ and $\alpha_0$ are free parameters, or with an exponential function with only three free parameters[35], also commonly used in asteroid works[29,36]. Because of the small number of data points, none of these functions allow us to make a reliable extrapolation of the behaviour of the polarisation of comet 2I/Borisov in the negative branch, nor at large phase angles. However, Eq. (1) may be conveniently used to describe the polarimetric curve of comet Hale-Bopp. Figure 1 shows also the best-fits to the Hale-Bopp data obtained with filters that have effective wavelengths close to those of the $R_F$ and $I_F$ FORS2 filters (note, however, that some outliers have not been considered for the best-fit to the data).

We finally note that polarimetric measurements of comet 2I/Borisov taken in the phase angle range 12.5–28°, using the

**Table 2 Polarimetric and photometric colours of 2I/Borisov as defined in the text.**

| $\alpha$ (°) | PSG$_{(557, 655\,nm)}$ %/100 nm | PSG$_{(655, 768\,nm)}$ %/100 nm | $(V–R)_{JC}$ | $(R–I)_{JC}$ |
|---|---|---|---|---|
| 28.48 | 0.42 ± 0.10 | 0.34 ± 0.16 | 0.41 ± 0.10 | 0.41 ± 0.09 |
| 27.56 | 0.74 ± 0.18 | – | – | – |
| 24.14 | – | 0.16 ± 0.37 | – | 0.45 ± 0.09 |
| 22.63 | – | 0.54 ± 0.25 | – | – |
| 18.06 | −0.23 ± 0.18 | – | – | – |

Advanced Camera for Survey/Wide Field Channel (ACS/WFC) of the Hubble Space Telescope (HST), have been recently presented in another study[37]. These observations, obtained in a filter F606W covering the wavelength range from 480 to 710 nm, have large uncertainties (about ten times higher than that of our measurement at the critical phase angle 28°). They do not allow to appreciate any similarity with comet Hale-Bopp, nor any difference in steepness when compared to the large majority of the other comets of our Solar System, but they suggest the lack of any region of locally higher positive polarisation surrounding the nucleus.

**Polarimetric spectral gradient**. Our measurements show that in the positive branch, the polarisation increases with wavelength; in the negative branch, we obtained only one measurement in the $V_F$ and $R_F$ filters, at a phase angle of 18°, and found that the absolute value of linear polarisation in the $R_F$ filter is slightly smaller than in the $V_F$ filter. Because our measurements were obtained with broadband filters, we cannot obtain a refined estimate of the polarimetric spectral gradient (PSG); the significance of a comparison with other comets is limited by the use of different filters, and also by the fact that polarimetric measurements of different comets are generally obtained at different phase angle values, but we can approximate

$$\left. \frac{d|P_r(\lambda)|}{d\lambda} \right|_{\lambda=\lambda_0} \simeq \mathrm{PSG}_{(\lambda_1,\lambda_2)} = \frac{|P_r(\lambda_2)| - |P_r(\lambda_1)|}{\lambda_2 - \lambda_1} \quad \text{where} \, \lambda_0 = \frac{\lambda_1 + \lambda_2}{2} \, , \qquad (2)$$

where $|P_r|$ stands for the absolute value of $P_r$. According to this definition, PSG is positive when the fraction of linear polarisation increases with wavelength, regardless of its direction, although is not defined around zero. In Table 2 we report the polarimetric spectral gradients PSG$_{(\lambda_1,\lambda_2)}$ of comet 2I/Borisov for the pair $\lambda_1 = 557$ nm, $\lambda_2 = 655$ nm, PSG$_{(557,655\,nm)}$, and for the pair $\lambda_1 = 655$ nm, $\lambda_2 = 768$ nm, PSG$_{(655,768\,nm)}$, at various phase angles. In Fig. 2 we compare these PSGs with those of comet Hale-Bopp estimated using observations[38] with filters centred at 484.5 nm (with a 65-nm wide passband), at 620 nm (with a 60-nm wide passband), at 670 nm (with a 30-nm wide passband), and at 730 nm (with a 50-nm wide passband), having calculated the PSG for the pairs $\lambda_1 = 484.5$ nm, $\lambda_2 = 620$ nm, and for the pair $\lambda_1 = 670$ nm, $\lambda_2 = 730$ nm.

**Polarimetry along the tails**. Polarimetric profiles were measured along both the gas-tail direction (anti-sunward direction) and the direction in which the dust tail is oriented, half-way between the anti-sunward and the direction opposite to the heliocentric velocity vector. Figure 3 shows the unpolarised image of comet 2I/Borisov obtained on 25/12/2019, and Fig. 4 shows the corresponding intensity and polarimetry profiles in all three filters along the horizontal direction and along the anti-sunward direction. Along both directions the polarisation profiles appear generally constant out to 20,000 km from the nucleus, while beyond 40,000 km the $S/N$ becomes too low to extract any useful information. Consistent tail behaviour is seen in the December

and January data, while the presence of numerous background stars prevented us from reaching firm conclusions from the observations obtained in the negative branch in March 2020. The stability of the polarisation throughout the coma is a sign of homogeneous ejection of the material from the nucleus, that is, a sign of weak contribution of active areas (if any) to the coma formation. We note that this behaviour greatly differs from the structures observed in the polarimetric profiles and polarimetric images of Hale-Bopp with a few thousands km scale[38,39] and of other comets[40,41].

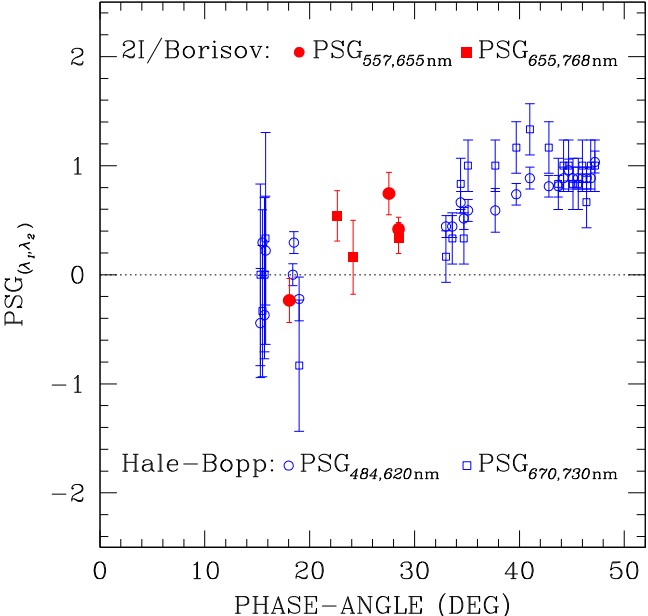

**Fig. 2 Polarimetric spectral gradients (PSG) of comets 2I/Borisov and Hale-Bopp.** For 2I/Borisov we used $\lambda_1 = 557$ nm, $\lambda_2 = 655$ nm (red solid circles) and $\lambda_1 = 655$ nm, $\lambda_2 = 768$ nm (red solid squares); for comet Hale-Bopp[38] we used $\lambda_1 = 484.5$ nm, $\lambda_2 = 620$ nm (blue empty circles) and $\lambda_1 = 670$ nm, $\lambda_2 = 730$ nm (blue empty squares). Error bars represent $1\sigma$ uncertainties, and are calculated from simple propagation of the polarimetric uncertainties that are estimated as explained in Methods subsection Polarimetry. We note that at phase angles values ≤21° the polarisation is directed along the scattering plane, at phase angle ≥21° it is directed along the perpendicular to the scattering plane.

**Photometric colours.** The optical colours derived from photometry, calculated as explained in Methods subsection Photometry, are presented in Table 2. They are in agreement with those found in previous studies[2,42,43], and broadly consistent with those of both short-period (Kuiper belt) and long-period (Oort cloud) comets, which indeed do not display any significant difference among themselves[4].

We note that the photometric colours of 2I/Borisov are much bluer than the ones for comet C/1995 O1 (Hale-Bopp), for which, for instance, it was reported $V - R = +0.71 \pm 0.07$ and $R - I = +1.03 \pm 0.13$ as an average value on 14–15 Aug 1995[44]. Interestingly, Hale-Bopp was observed to become bluer on 16 Aug 1995, reaching similar values to the ones observed for 2I/Borisov, suggesting that this shift in colour are explained by an outburst which ejected a large quantity of very small ice grains[44]. Another study[17] also indicates that the blue colour was typical for the spiral structures in the Hale-Bopp coma, which decreased the average coma colour.

The photometric time-variability of the photocentre during an observing series in each filter (~15–20 min) is smaller than the standard error of each individual measurement, and no clear trend could be discerned. This could be caused by a nearly spherical shape for the nucleus or a slow rotation, but the most plausible cause is that the signal contribution from the coma prevents direct observation of the nucleus rotation[45].

## Discussion

The photometric data obtained in our and other investigations have large uncertainties, and are compatible with very different classes of small bodies of our Solar System, including short and long-period comets, Centaurs, Jupiter Trojans and some main belt asteroids[4], and the same is true for the reflectance spectra. In summary, photometric data do not point to 2I/Borisov as an object with distinctive characteristics. Instead, the polarimetric characteristics of 2I/Borisov are more suggestive of its interstellar origin.

Several studies[46,47] have divided comets into two polarimetric classes: low- and high-polarisation comets. This different behaviour is usually ascribed to a very different dust-to-gas ratio in the coma[8]. Since gas contamination dilutes the polarisation produced by the dust scattering, the higher dust-to-gas ratio, the higher the comet polarisation. However, the gas contamination is a consequence of the difference in the intrinsic properties of the dust particles. In low-polarisation comets, the particles are bigger and more compact than in higher polarisation comets, hence they are concentrated near the nucleus, and the coma is dominated by gas. In high-polarisation comets, the particles are smaller and more

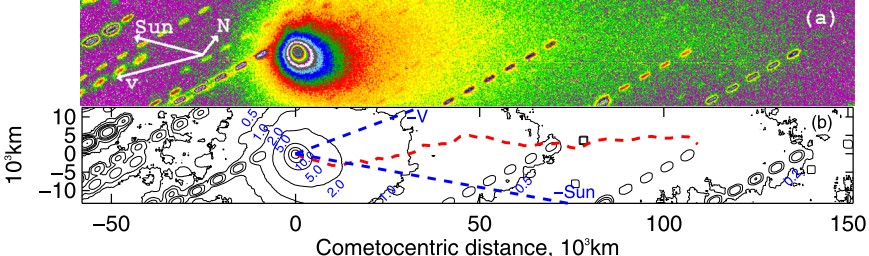

**Fig. 3 Imaging of comet 2I/Borisov.** Panel **a**: false colour image of 2I/Borisov; panel **b**: isophotes in e⁻s⁻¹. Imaging was obtained by stacking all polarimetric frames obtained with the $R_F$ filter on 15/12/2019. The colours help to highlight the rapid spatial variation of the flux; the flux distribution may be quantitatively appreciated with the help of the isophotes of panel **b** (see also panels **a** and **e** of Fig. 4). Dust tail is extended for ~3 arcmin approximately along the horizontal direction of the CCD (the 22″ wide Wollaston strips of the instruments are oriented 128° North to East; the direction of the heliocentric velocity vector **v** is oriented ~143° and the sunward direction is at ~112°). The dashed tracks are due to trailing stars. The red dotted line of panel **b** traces the comet tail, which at shorter distance from the photocentre is directed along the anti-solar direction, and at larger distances lies in between the anti-solar and anti-velocity direction. Spatial scale of panel **a** is the same as of panel **b**.

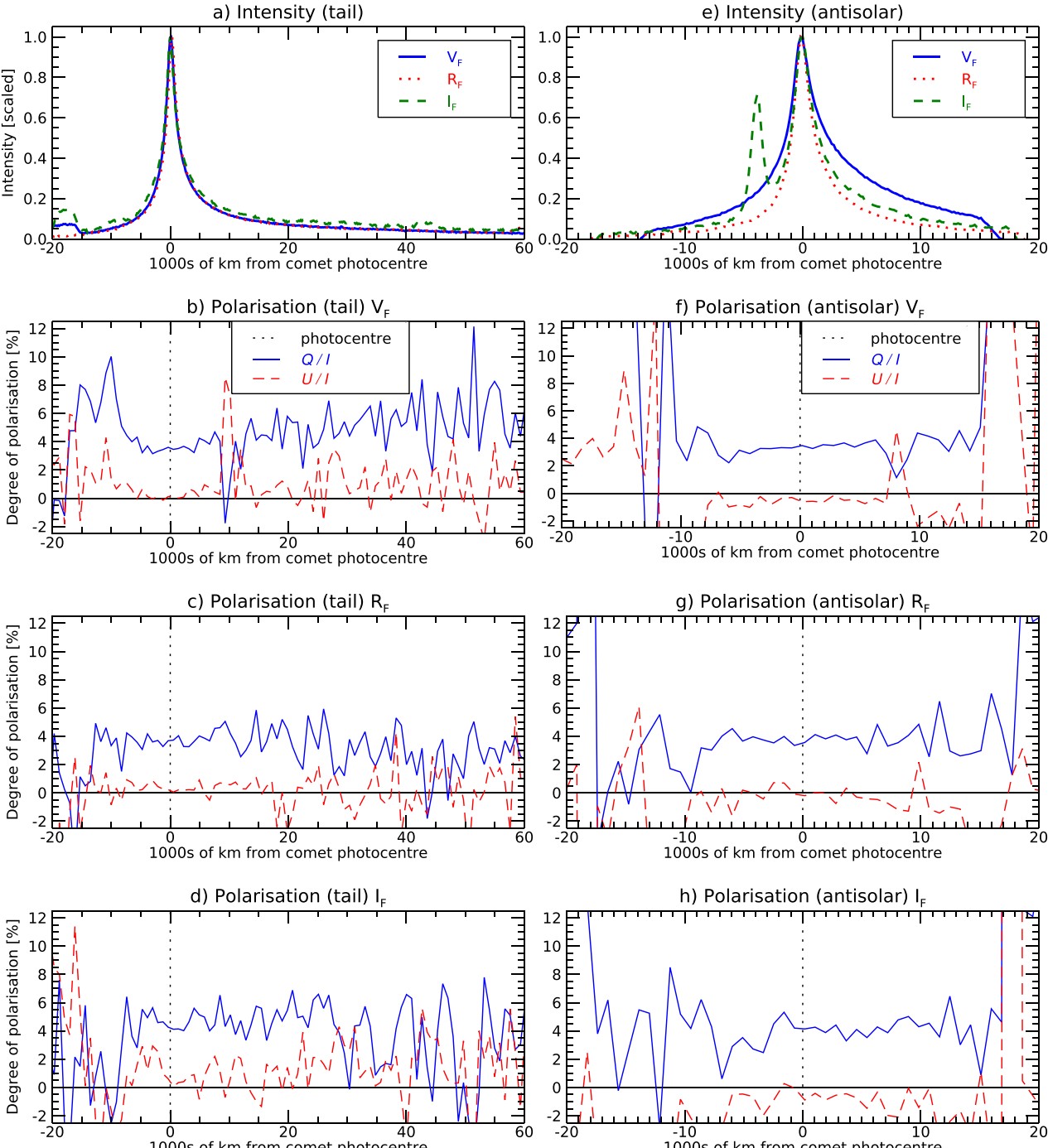

**Fig. 4 Normalised intensity profiles and polarimetric profiles of 2I/Borisov.** We have considered the observations in all filters $V_F$, $R_F$ and $I_F$ along the dust tail in a direction of the extended dust tail, which is intermediate between the anti-sunward direction and the direction opposite to the heliocentric velocity (panels **a**–**d**), and along the anti-sunward direction (panels **e**–**h**). The photometric intensities are affected by occasional contamination from background stars. $P_r$ (represented with blue solid lines) is constant along both directions, while, as expected from symmetry, $U/I$ (represented with red dashed lines) is close to zero throughout. The spatial dispersion is 177 km/pixel (1 pixel = 0.126″), but to increase the signal-noise ($S/N$), polarimetric data were rebinned to 5 pixels (880 km).

porous, capable of populating the coma and creating a high dust-to-gas ratio environment. This picture is supported by the difference in orbital characteristics of the two groups: comets with a gas-dominated coma are old comets with small perihelia and aphelia, hence strongly affected by solar radiation and solar wind, whereas dusty comets have larger perihelia and aphelia and experience less weathering. New comets, not yet processed by solar wind, are expected to be dusty, and to show high polarisation. Other interpretations invoke a different dust composition[48], or explain high-polarisation comets with a higher level of activity[49] (high-polarisation comets would have for instance more jets with small particles than inactive comets, and in the jets there are smaller particles and thus, higher polarisation). However, both these interpretations fail to explain the fact

that the observed polarisation tends to decrease with the distance from the nucleus.

There exists a third polarisation class[50], so far represented by a single comet, C/1995 O1 (Hale-Bopp). This comet was observed up to phase angle ≃50°, and showed an increase of polarisation with phase angle noticeably exceeding the other high-polarisation comets. At the epoch of its measurement, the polarimetric curve of comet C/1995 O1 (Hale-Bopp) was steeper and higher than that of any other comet previously observed. Hale-Bopp is a long-period comet, having a period currently >2500 ys, as determined during its perihelion passage in 1995–1997. The peculiar positive polarisation behaviour for Hale-Bopp was explained by domination of small particles in its coma, with an estimated behaviour approaching the Rayleigh scattering regime[51]. This was consistent with the disappearance of the negative polarisation branch in the near-IR[52], but especially with thermal IR spectroscopy, which showed a strong silicate feature[53] typical for small dust particles[54]. The Deuterium/Hydrogen (D/H) ratio in the $H_2O$ of comet Hale-Bopp inferred from spectra is $>10^{-4}$, about ten times that of the value commonly assumed for the proto-solar nebula[55]. This was interpreted as diagnostic of a comet that originated in the outer Solar System, at temperatures ~30 ± 10 K, an astrophysical environment probably distinct but not too dissimilar from that of the interstellar medium[55].

Figure 1 shows that, within the range of phase angles covered by our observations, 2I/Borisov has a polarimetric curve remarkably similar to the unique curve of comet Hale-Bopp, and different from that of any other comet: the probability the two measurements around phase angle 27–28° in the $R_F$ filter could be the outliers of an otherwise normal behaviour of Solar-System comets is $<10^{-2}$ (see Methods subsection Statistical tests).

Figure 2 shows also similarities in the polarimetric spectral gradient of the two comets. In fact, the PSG characteristics of Fig. 2 are not so rare; most of the comets display, like comets Hale-Bopp and 2I/Borisov do, a positive PSG in the positive branch. In the negative branch, the PSG of comets is also generally positive at visible wavelengths[56,57], although data available are still limited, and often obtained at low significance level. A negative PSG in the positive branch was found only for a small number of comets belonging predominantly to the short-period and/or Jupiter family classes, which in this respect seem to exhibit a clearly different behaviour with respect to long-period comets[8].

The similarity between the polarimetric properties of the two comets must depend upon the microscopic structure and composition of the aggregates, and not on their macroscopic characteristics, as the two comets are quite different in size: the analysis of the photometric profile of the inner coma suggests that comet Hale-Bopp belongs to the class of giant comets, with the diameter of the nucleus being estimated between 20 and 35 km[58], while 2I/Borisov's nucleus size is ≤0.4 km[43,59]. The close similarity between the polarimetric behaviour of the comet 2I/Borisov and Hale-Bopp suggests that, whatever astrophysical environment in which comet 2I/Borisov originated in, such environment had properties which led to the formation of a body bearing significant analogies with those accreted in the outer regions of our Solar System, a remarkable result on its own. This similarity could also suggest that the dust particles of 2I/Borisov are small, like those of Hale-Bopp. We are not aware of any measurements of comet 2I/Borisov in the thermal infra-red that could be used to set a constraint on particle size; however, close to perihelion, 2I/Borisov exhibited a NIR spectrum with a negative slope, which was explained as an increase in water ice and/or decrease in dust size[60]. HST imaging of the comet was modelled assuming coma particle size ~100 μm[61], and other models have found particles of millimetre size particles[62], in contrast with our claim. However, what these models derive is actually the ratio between solar

radiation pressure and gravity force $\beta$ which in turns does not have a one-to-one relationship with the particle or aggregate size. In fact, the best-fits values for $\beta$ could either correspond to large radii, or to dust-grains with radius ≤ 0.1 μm[63], or to aggregates as small as 0.2 μm[64]. We note that extremely high polarisation values were measured for comet C/1999 S4 (LINEAR) after it started to break apart[41]; since these data were obtained when the comet was seen in a phase angle range very different than our observations of comet 2I/Borisov, it is difficult to make a direct comparison; however, it is reasonable to hypothesise that comet C/1999 S4 was releasing small particles during this event, which were responsible for the high degree of the observed polarisation. We finally note that if we assumed that the same relationship between albedo and polarisation found for asteroids also holds for comets, our observations would indicate that also 2I/Borisov has a low geometric albedo, a property shared indeed by most comets of our Solar System.

When discovered, Hale-Bopp was among the brightest comets ever seen, and displayed cometary activity at large heliocentric distances, a fact interpreted to indicate a high-volatile content. Its polarimetric images showed clear structures, revealing the presence of jets and arcs[39]. By contrast, at the time of our observations, comet 2I/Borisov was polarimetrically homogeneous, showing no sign of active areas contributing to the coma formation. Prior to its recent perihelion passage, comet Hale-Bopp probably was near the Sun at least once, and possibly only once, ~2250 BC[65]; at the time of that first approach, the original material was removed from the surface and active areas were open[66], hence Hale-Bopp could manifest activity during its recent perihelion passage. Comet 2I/Borisov instead, most likely never passed close to the Sun or any other star, and may represent the first truly pristine comet that has ever been observed.

## Methods

**Polarimetry**. Polarimetry was obtained using the beam-swapping technique[67], setting the retarder waveplate at the eight position angles 0°, 22.5°, …, 157.5°, although in one case (the measurement with the $R_F$ filter on 2020-01-08) some exposures had to be discarded because of the presence of background objects too close to the comet photocentre, and in one case ($I_F$ filter on 2019-12-25) we deliberately used only the four positions 0°, 22.5°, 45°, 67.5° of the retarder waveplate to save overhead time. The reduced Stokes parameters $X/I$ ($X = Q, U$) were calculated as[67]

$$\frac{X}{I} = \frac{1}{2N} \sum_{j=1}^{N} \left[ \left( \frac{f^{\parallel} - f^{\perp}}{f^{\parallel} + f^{\perp}} \right)_{\alpha_j} - \left( \frac{f^{\parallel} - f^{\perp}}{f^{\parallel} + f^{\perp}} \right)_{\alpha_j + 45°} \right]. \quad (3)$$

where $f_{\alpha}^{\parallel}$ and $f_{\alpha}^{\perp}$ are the fluxes measured in the parallel and perpendicular beams, respectively, with the retarder waveplate at the position angle $\alpha$; for $X = Q$, $\alpha_j \in \{0°, 90°\}$; for $X = U$, $\alpha_j \in \{22.5°, 112.5°\}$; $N$ is the number of $\alpha_j$ values of the observing series ($N = 2$ for most of our observations). This double difference method is shown to give practically the same results as the double ratio method[67]. The uncertainties are given by

$$\sigma_{X/I}^2 = \frac{1}{(2N)^2} \sum_{j=1}^{N} \left[ \left( g(\alpha_j) \right)^2 + \left( g(\alpha_j + 45°) \right)^2 \right] \quad (4)$$

where

$$g^2(\alpha) = \left( \frac{2f^{\parallel}f^{\perp}}{(f^{\parallel} + f^{\perp})^2} \right)^2 \left( \frac{(\sigma_{f^{\parallel}})^2}{(f^{\parallel})^2} + \frac{(\sigma_{f^{\perp}})^2}{(f^{\perp})^2} \right)_{\alpha} \quad (5)$$

and $\sigma_{f^{\parallel}}$, $\sigma_{f^{\perp}}$ are the uncertainties of the fluxes $f^{\parallel}$ and $f^{\perp}$, respectively. Sky background was generally calculated in a region close to the comet but with little apparent contamination from the coma. FORS2 instrumental polarisation around the centre of the field of view is ≤0.03%, hence negligible in the context of our observations[68], but background polarisation must be estimated within ~1 arcmin from the source to avoid instrument polarisation that becomes significant at the edge of the field of view[69]. The reduced Stokes parameters are reported adopting as a reference direction the perpendicular to the great circle passing through the object and the Sun, using the formula

$$\frac{Q}{I} = \cos\left(2(\chi + \epsilon + \Phi + \pi/2)\right) \frac{Q'}{I} + \sin\left(2(\chi + \epsilon + \Phi + \pi/2)\right) \frac{U'}{I} \quad (6)$$

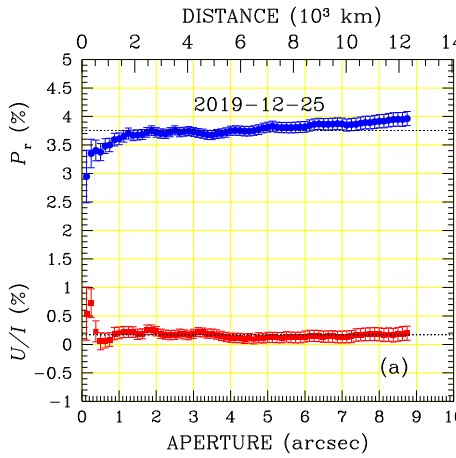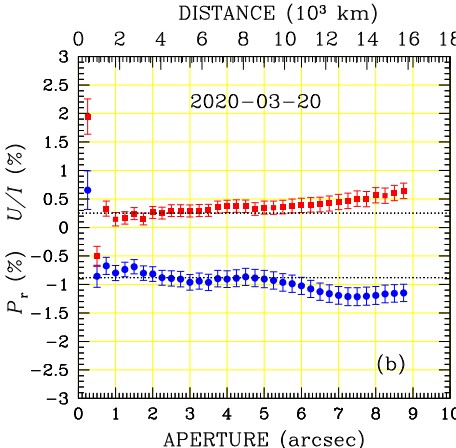

**Fig. 5 Aperture polarimetry of 2I/Borisov in the $R_F$ filter.** Blue filled circles refer to $P_r$ and red filled squares refer to $U/I$. The dotted black solid lines show the values adopted in Table 1. Panel **a** refers to data obtained on 2019-12-25, when polarisation was directed along the direction perpendicular to the scattering plane. Panel **b** refers to observations obtained on 2020-03-20, when polarisation was directed along the scattering plane; for these observations, CCD readout was rebinned 2 × 2, which explains the less refined sampling of the growth curve (compared to **a**). Reduced Stokes parameters are computed in circles of radius given in the abscissa. The Wollaston strip is 22″ wide, therefore aperture radius cannot be larger than 11″. Practically, the aperture limit is determined by the contamination of background stars (see Fig. 3). Error bars are calculated as explained in Methods subsection Polarimetry.

$$\frac{U}{I} = -\sin\left(2(\chi + \epsilon + \Phi + \pi/2)\right)\frac{Q'}{I} + \cos\left(2(\chi + \epsilon + \Phi + \pi/2)\right)\frac{U'}{I} \quad (7)$$

where $Q'$ and $U'$ are the Stokes parameters measured in the instrument reference system, $\chi$ is the instrument position angle (counted counterclockwise from North to East) at the time of the observations, $\epsilon$ is an angle, filter dependent, that is introduced to correct for the chromatism of the retarder waveplate[67], $\Phi$ is the angle between the direction Object-North Pole and the direction Object-Sun. This angle can be calculated applying the four parts formula to the spherical triangle defined by the object with coordinates ($\alpha_T, \delta_T$), the Sun, with coordinates ($\alpha_\odot, \delta_\odot$) and the North celestial pole[70]:

$$\sin\delta_T\cos(\alpha_\odot - \alpha_T) = \cos(\delta_T)\tan(\delta_\odot) - \sin(\alpha_\odot - \alpha_T)\frac{1}{\tan(\Phi)} \ . \quad (8)$$

$Q/I$ estimated from Eqs. (6)–(7) represents the flux perpendicular to the plane Sun-object-Earth (the scattering plane) minus the flux parallel to that plane, divided by the sum of these fluxes. For aperture polarimetry, Stokes parameters were calculated from fluxes measured in apertures up to ~9″ size at one or two pixel (=0.125/0.25″) increments, and the adopted polarisation value is determined by the plateau observed in the growth curve, as illustrated in Fig. 5. As a further quality check, we have also calculated the null parameters $N_Q$ and $N_U$ that are the difference between the corresponding reduced Stokes parameters calculated from consecutive pairs of measurements[67] using the formulas

$$N_X = \frac{1}{2N}\sum_{j=1}^{N}(-1)^{(j-1)}\left[\left(\frac{f^\parallel - f^\perp}{f^\parallel + f^\perp}\right)_{\alpha_j} - \left(\frac{f^\parallel - f^\perp}{f^\parallel + f^\perp}\right)_{\alpha_j + 45°}\right] \ . \quad (9)$$

Null parameters are expected to have a Gaussian distribution centred about 0 with the same $\sigma$ given by Eq. (4), and their deviations from zero would flag the possible presence of systematics effects. Null parameters were found consistent with zero for all datasets except those obtained on 2020-02-06, where they deviate from zero by $\simeq 5\sigma$ for aperture radius > 1″. The presence of systematic effects, also suggested by the deviation from zero of $U/I$ in the $I_F$ filter, is likely due to the bad seeing conditions under which the observations were carried out (the observations were actually repeated at a later epoch), joined to a particularly crowded background. For this epoch, polarimetry was measured within a 1″ wide aperture.

In addition to aperture polarimetry described above, we also employed a slit polarimetry method[25] which allows us to measure polarisation along the tail, or any other chosen direction. A slit of adjustable width was chosen around the comet nucleus and aligned with the comet tail or other features. In this way, most of the background sources can be avoided, improving the quality of the final results, although some comet signal is lost in the process. Some artefacts remain, best seen in the photometric plots of $I_F$ filter in Fig. 4. A similar method was used in the past to obtain radial profiles of the tail and jets of comet 67P/Churyumov-Gerasimenko[71]. The width of the slit, centred on the comet photocentre, was between 36 and 42 pixels (4.5–5.25″) across the dust tail (Fig. 4a–d) and between 18 and 20 pixels (2.25–2.5″) across the anti-sunward direction (Fig. 4e–h).

**Photometry**. We measured the comet coma brightness using circular apertures of 2.5″, which were large enough to include the total flux of trailed stars for which the magnitude was known through Gaia[72] observations. As a drawback for using such a large aperture size, most of the frames had to be discarded due to the crowded stellar background which contaminated the photometry of the comet. In the end, we obtained reliable photometry only from data obtained during the night of 2019-12-25 for each of the FORS2 filters used ($V_F$, $R_F$ and $I_F$), and during the night of 2020-02-06 for the images gathered with $R_F$ and $I_F$ filters.

In order to calculate the dust optical colours in the Johnson-Cousins UBVRI system, we calculated an empirical transformation between FORS2 filters and the former system (private communications with C. Jordi). We integrated the spectra for Johnson-Cousins R, V and I filters, as well as for $V_F$, $R_F$ and $I_F$ filters using a virtual star with $G = 15$ without adding any interstellar extinction. The choice of the magnitude selected is irrelevant since we are only interested in the relative difference between values. The relation between filters resulted to be almost perfectly linear ($r = 0.999$) and therefore the transformation can be expressed in the form of the linear equations

$$(V - R)_{JC} = -0.0171 + 1.1101\,(V - R)_F \quad (10)$$

$$(R - I)_{JC} = 0.0315 + 0.9558\,(R - I)_F \quad (11)$$

where $(V - R)_{JC}$ and $(R - I)_{JC}$ are the optical colours expressed in the Johnson-Cousins system, whereas $(V - R)_F$ and $(R - I)_F$ are the colours measured with the corresponding FORS2 filters.

**Statistical tests**. Statistical prediction limits for the observations at significance level $(1 - q) \times 100\%$ may be expressed as

$$\text{PL}_{q \times 100}(\alpha) = P_r(\alpha, \hat{p}) \pm t_{n-4}(q/2)\sqrt{\frac{\sum_i^N (Y_i - P_r(\alpha_i; \hat{p}))^2}{n - 4}}\sqrt{1 + J^T(\alpha)C_p J(\alpha)} \quad (12)$$

where $N$ is the sample size, $Y_i$ is the measurement at $\alpha_i$, $p = [b, c1, c2, \alpha_0]$ is the parameter vector, $\hat{p}$ and $C_p$ the parameter estimates and associated covariance from the fit, $J = \nabla_p Pr(\alpha; p)$ evaluated at $p = \hat{p}$ and $t^{n-4}(q/2)$ is the $q/2$ percentage point of Student's distribution with $n - 4$ degrees of freedom. For the asymptotic case $n = \infty$ we have $t(0.05) = 1.645$ and $t(0.025) = 1.960$. To quantify the disagreement between 2I/Borisov and the other comets, we fitted the ensemble cometary data to Eq. (1) using nonlinear least squares minimisation, subtracted the fit from the data and compared the predictive confidence limits (12) on the observations to the Borisov measurements. The result for the $R_F$-filter and $I_F$-filter measurements separately is shown in Fig. 6. Here we considered only data with available observational uncertainties and $0 < \alpha < 60°$, yielding sets with $N = 393$ ($R_F$ filter) and $N = 113$ ($I_F$ filter) respectively. We then discarded the 5% of each dataset with the highest uncertainties. Our two $R_F$-filter measurements of 2I/Borisov at $\alpha = 27.5°$ and $\alpha = 28.6°$ (red points, panel a) lie between $PL_{05}$ and $PL_{10}$, therefore the probability $P$ that both $R_F$-filter measurements follow the same phase-polarisation behaviour as the other comets is $2.5 \times 10^{-3} < P_R < 10^{-2}$. Our $I_F$-filter measurement of 2I/Borisov at $\alpha = 28.5°$ (panel b) lies between the same two predictive contours and therefore $P_I < 10^{-1}$. Though the $R_F$ and $I_F$ measurements must be correlated to some degree, in any case $P_{R_F, I_F} < 10^{-2}$.

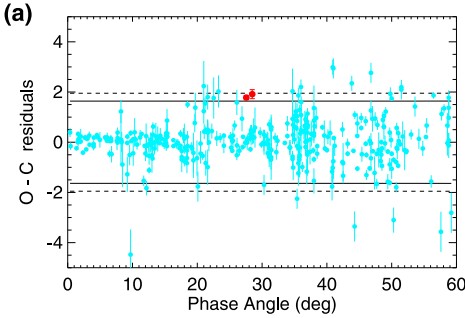
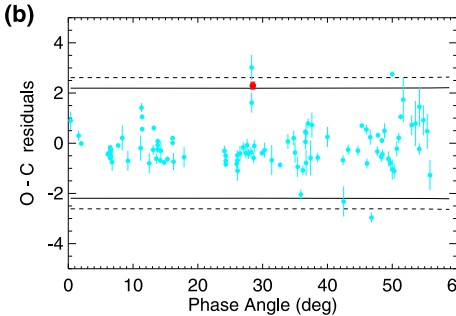

**Fig. 6 Residuals of cometary polarisation measurements.** Residuals of all comet polarisation measurements (cyan symbols) except those of comet Hale-Bopp fitted to Eq. (1), compared to our data of comet 2I/Borisov obtained at the largest phase angles (red symbols). Panel **a** refers to observations obtained in the $R_F$ filter and panel **b** to the observations obtained in the $I_F$ filter. The continuous and dashed curves correspond to statistical prediction bands for 90% and 95% confidence respectively. Errorbars are calculated as explained in Methods subsection Polarimetry.

## Data availability

FORS2 data are available in the ESO archive (archive.eso.org) under programme ID 2104.C-5003.

## Code availability

IRAF is the Image Reduction and Analysis Facility, a general-purpose software system for the reduction and analysis of astronomical data. It is written and supported by the National Optical Astronomy Observatories (NOAO) in Tucson, Arizona, USA. NOAO is operated by the Association of Universities for Research in Astronomy (AURA) under cooperative agreement with the National Science Foundation. The code is available from http://iraf.noao.edu/. A simple FORTRAN code was used to read the standard IRAF output and combine the fluxes to obtain aperture polarimetry as explained in Methods subsection Polarimetry. Three custom-made IDL (Interactive Data Language, version 8.7, by Harris Geospatial Solutions, Boulder, Colorado, USA) routines were also utilised for the computation of the polarimetric profiles. These codes are available upon reasonable request to the authors. Astrometrica (www.astrometrica.at) was used for the automatic identification of reference stars for the photometric measurements.

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

## Acknowledgements

This work is based on observations made with ESO Telescopes at the La Silla Paranal Observatory thanks to Director Discretionary Time under programme ID 2104.C-5003. The work of T.S.-R. was carried out through grant APOSTD/2019/046 by Generalitat Valenciana (Spain). This work was supported by the MINECO (Spanish Ministry of Economy) through grant RTI2018-095076-B-C21 (MINECO/FEDER, UE). We would like to thank Carme Jordi (ICCUB-IEEC) for her kindly contribution on the calculation of the empirical colour transformation between photometric systems.

## Author contributions

S.B. contributed to the writing of the proposal, prepared the observations, reduced polarimetric data, contributed to all aspects of the discussion and of the paper writing and led the project; A.C. contributed to the writing of the proposal, to the data interpretation and to the writing of the paper; L.K. contributed to the data interpretation and to the writing of the paper; R.N. contributed to the polarimetric data reduction (in particular for Fig. 4) and to the writing of Methods subsection Polarimetry; T.S.-R. reduced the photometric data, calculated the optical colours and contributed to the writing of Methods subsection Photometry; G.B. contributed to all aspects of the polarimetric data reduction, to the tracing of the gas and dust tails, and to the discussion; A.A.C. performed the statistical test and wrote Methods subsection Statistical tests; M.D. contributed to the review of the literature data; P.B. participated in the preparation of the proposal and in the discussion of the results. All authors reviewed the manuscript.

## Competing interests

The authors declare no competing interests.
