## [Peer Review File · Nature Communications]

REVIEWER COMMENTS

Reviewer #1 (Remarks to the Author):

This is a very interesting paper about polarimetric properties of interstellar comet 2I/Borisov. This amazing object has been the subject of many studies, as it entered our Solar System by the end of 2019 and has been close enough to the Earth for a significant period of time so that detailed observations were possible. Polarimetric observations provide very useful information on the physical characteristics of cometary dust grains, that are quite difficult to obtain by other techniques. So far, and with the exception of a DPS2020 contribution by Zhang et al., this is the very first study of comet 2I/Borisov using this technique. All in all this paper presents original results, data have been presented and processed properly, results are reproducible and error analysis is correct. I recommend this paper for publication only after minor corrections.

Here are my comments:

1. Maybe the authors would like to cite the results presented at the DPS2020 by Q. Zhang, on polarimetric imaging of 2I/Borisov with the HST and the ACS/WFC. Polarimetry was collected through the F606W filter, covering a range of phase angles from 12 to 29 degrees, as the comet receded from the Sun. They found that polarization of 2I/Borisov is broadly consistent with that of most previously characterized comets. Please have a look at the attached figure, which is a plot of the degree of polarization vs. phase angle presented by Zhang et al. The orange points correspond to data extracted from the literature, exactly from the same source as the authors used for this paper. Zhang et al. also did a polarization map, with no definitive polarimetric structures resolved.
2. In the Discussion section, the authors comment on the NIR blue spectrum of Borisov, "which was interpreted as evidence for the presence of small particles". I suggest the authors to soften a bit this statement. The fact that the comet presented a negative, blue reflectance spectrum in the NIR when observed close to perihelion, and I cite Lee et al. (2020) "can be caused by an increased abundance of pure water ice or hydrated minerals, which is likely as the cometary activity is transitioning from CO-dominant volatile to H₂O-dominant volatile, and the comet is close to the perihelion". Actually, Lee et al. only rise the possibility of a decrease of dust size in their abstract, where they say "Possible causes of the decreasing slope are an increase in water ice and/or decrease in dust size".
3. A very recent paper from Kim et al. (2020) presents results on images of 2I/Borisov obtained by the HST between October 2019 and January 2020. The authors modeled the dust, finding that, while the coma particle size from October 2019 observations was around 100 μm (consistent with earlier

reports), they were not able to model the direction of the dust coma in the November and December observations using the same size, indicating that the coma particle size could not be characterized by a single value. This suggests that particles are ejected anisotropically. They also observed an asymmetry in the coma in the January 2020 observations, that could only be reproduced by assuming particles of 1 mm. Millimeter sized particles were independently identified in a model by Cremonese et al. (2020). I suggest the authors to incorporate these results into their discussion.

4. Are there other similarities in terms of size, colors, etc., between comet 2I/Borisov and Hale-Bopp? I would like the authors to expand a bit their comparison of these two objects. In addition, I do not entirely understand the reason to present a comparison with Bennu. Is the polarization curve of Bennu different from that of other low-albedo asteroids? The authors only say that it shows an "unusually steep phase-polarization curve". They should expand this argument, and if so, if Bennu behaves different from other asteroids, even from other "active asteroids", then its observed activity can be used as a potential explanation of this behavior. Please also consider that the level of activity observed in Bennu is very far away for that observed for comets.

Reviewer #2 (Remarks to the Author):

REVIEW OF MANUSCRIPT:

S. Bagnulo et al., Unusual polarimetric properties for interstellar comet 2I/Borisov, submitted to Nature Communications, manuscript NCOMMS-20-43192

The manuscript reports the degree of linear polarization and polarimetric color for the interstellar comet 2I/Borisov observed with the ESO/VLT FORS2 instrument. The observations suggest to place the object in a unique category of comets, with closest apparent similarity to Comet C/1995 O1 (Hale Bopp). The uniqueness is explained by the pristine material of the comet, providing us with information about the presumably distant place of origin for the comet.

There are a small number of data points collected at the solar phase angles of 18-28 degrees. Additional observations at smaller phase angles in spring 2020 were lost due to the restrictions set by the Covid-19 pandemic. The data points collected allow the retrieval of the polarimetric slope at the inversion angle, but the scarce data make it impossible to predict the polarimetric behavior at smaller and larger phase angles with any of the existing empirical models. The manuscript can become publishable after revision according to the comments below.

The authors' claim that the polarimetric data are unusual, as stated already in the title, should be quantified by a statistical test. This must be done due to the small number of observed data points (Table 1, Fig. 1).

There is no consensus in the planetary science community about the hypothesis that there would be two classes of comets, that is, low-polarization and high-polarization comets. The authors should offer a more balanced view about the models for cometary polarization.

No physical modeling is carried out for the observations in the manuscript. Thus, reviews of existing models are called for. It turns out that some cometary polarization models are reviewed but some are not. For example, it would be necessary to widen the selection by reviewing and analyzing the following modeling works:

1) Zubko et al. (2015), Characteristics of dust in Comet C/1995 O1 (Hale-Bopp) inferred with polarimetry, The Sixth Moscow Solar System Symposium, Moscow, Russia, pp. 68-70, 2015, openly available, and references in the paper.

2) Markkanen et al. (2018), Interpretation of the phase functions measured by the OSIRIS instrument for Comet 67P/Churyumov-Gerasimenko. *Astrophys. J. Lett.* 868, L16.

The former assesses Hale-Bopp that is clearly important for the present study and the latter assesses the Rosetta mission target comet, for which very large coma particles were measured by Rosetta in situ.

Additionally, there are two minor points to be accounted for:

1) Currently, both "solar system" and "Solar System" appear frequently in the text.

2) "Trans-Neptunian" should rather read "transneptunian"

(cf. "extraterrestrial").

Karri Muinonen Helsinki,

November 16, 2020

Reviewer #3 (Remarks to the Author):

This paper, submitted to Nature Communication, presents unique results about polarimetric measurements on interstellar comet 2I/Borisov and their interpretation.

It is well structured and pleasantly written by experts on polarimetric observations of surfaces of small bodies and of dust particles ejected by cometary nuclei, and certainly deserves publication in Nature Communications.

While the observations and their interpretation are remarkable, it would be better to:

1. Provide some background on polarimetric observations and their significance, for readers not so much familiar with such topics.
2. Tentatively find more information through Figure 3.
3. Extend the discussion, in terms of possible contamination of the polarization (induced by solar light scattering on dust particles) by gaseous emissions. Also, update the discussion, with complementary observations of a new comet presenting, as C/1995 O1 Hale Bopp, a very high polarization, and possibly with references to a couple of recent papers about cometary dust properties.

1. Introduction (2nd paragraph)

1.1. While comets and asteroids are mentioned, it could be better to explain that the information provided by polarization induced by scattering on surfaces (asteroids or cometary nuclei) is easier to interpret than the polarization induced by the scattering on huge dust clouds, such as cometary comae or tails. In the latter case, the signal corresponds to different conditions along the line-of-sight, such as the distance to the nucleus; the dust particles may exhibit quite different properties in dust jets and possibly in innermost comae (e.g., Hadamcik & Levasseur-Regourd, Icarus, 2003; Kiselev et al. review, In Polarimetry of stars and planetary systems, Kolokolova et al. Eds, CUP, 2015).

1.2. It could also be explained that polarimetric observations, complemented by numerical and experimental simulations, may provide information on the size and size distribution, on the morphology and porosity, and on the complex refractive indices, whence clues to the composition (again in Polarimetry of stars and planetary systems, 2015, reviews by, e.g., Mishchenko; Levasseur-Regourd et al.; Cellino et al.; Kiselev et al.).

2. Polarimetry along the tails

2.1. Nice Figure 3 would be improved by indications of the orientations of the dust and gas tails.

2.2. It would be of interest to provide a B/W figure with iso-contours of the coma (possibly removing the right part of Fig. 3), which could enhance some asymmetries.

2.3. The orientation of the dust trail (along the trajectory) could be searched, in the (unlikely) case of some heterogeneities suggesting the presence of large fragments.

3. Observations and their discussion

3.1. The observations are obtained for values of the phase angle between 18° and 29°, that is to say in the so-called inversion region. Results on the inversion angle at given wavelengths could be compared with those obtained on asteroids and other media.

3.2. The errors bars of the values obtained through the (quite large) R and I filters should be discussed, taking into account the possible contamination by gaseous emissions, which can only be avoided through narrow interference filters dedicated to cometary dust observations. Indeed, most cometary dust polarimetric observations are done with narrow interference filters, as typically provided by ESA for Churyumov-Gerasimenko observations (see, e.g. Hadamcik et al., MNRAS 2017).

3.3. The bandwidths of all filters mentioned in Table 1 should be listed in the Table or its caption.

4. Comparison of polarimetric properties of 2I/Borisov with those of other comets

An significant result is missing in this part of the discussion, since comet C/1995 O1 Hale-Bopp is not the single one to have shown a very high polarization. At least an other comet, C/1999 S4 LINEAR, has been found to present a high polarization (above 100° phase angle). It can be noticed on Fig. 4 of Hadamcik & Levasseur-Regourd, Icarus 2003, i.e. reference # 30 in the paper presently reviewed.

5. Minor comments

- Grammar and spelling possibly needing to be checked. E.g., “Measurements of cometary linear polarisation provideS information about...”

- Figure 1. While the most visible data points and fits correspond to Bennu and Hale-Bopp (in dark blue), the new and fascinating data points related to 2I/Borisov should be more conspicuous.
- Figure 2. The colour code is completely different from the one used in Fig. 1. It may be delicate for the reader, since those two figures could be close to one another in the paper.
- It would be better to follow the terminology now accepted for cometary dust particles (Güttler et al., A&A 630 2019). The wording “grains” should be used for the irregular and compact monomers, i.e. tiny building blocks, of the dust particles. The dust particles are irregular and fractal “aggregates” or “agglomerates”, most likely with a wide range of porosities (e.g. Mannel et al., also within A&A 630, 2019). The wording “particulates” could be avoided.

Anny-Chantal Levasseur-Regourd, Nov. 2020

PS for the authors. There is quite possibly a reason why 2I/Borisov and C/1995 O1 Hale-Bopp have similar polarimetric trends.

It is an other story, out of the scope of this paper, which I would be happy to discuss carefully with the authors in a near future.

We thank all the referees for their very useful comments and criticisms. Here come our answers, and the descriptions on how we have implemented their suggestions. Below we quote their reports with italic fonts. In addition to the changes required to meet the referee comments, we have also changed the definition of the polarimetric spectral gradient and used the absolute value of the polarisation.

REFeree 1

This is a very interesting paper about polarimetric properties of interstellar comet 2I/Borisov. This amazing object has been the subject of many studies, as it entered our Solar System by the end of 2019 and has been close enough to the Earth for a significant period of time so that detailed observations were possible. Polarimetric observations provide very useful information on the physical characteristics of cometary dust grains, that are quite difficult to obtain by other techniques. So far, and with the exception of a DPS2020 contribution by Zhang et al., this is the very first study of comet 2I/Borisov using this technique. All in all this paper presents original results, data have been presented and processed properly, results are reproducible and error analysis is correct. I recommend this paper for publication only after minor corrections.

Maybe the authors would like to cite the results presented at the DPS2020 by Q. Zhang, on polarimetric imaging of 2I/Borisov with the HST and the ACS/WFC. Polarimetry was collected through the F606W filter, covering a range of phase angles from 12 to 29 degrees, as the comet receded from the Sun. They found that polarization of 2I/Borisov is broadly consistent with that of most previously characterized comets. Please have a look at the attached figure, which is a plot of the degree of polarization vs. phase angle presented by Zhang et al. The orange points correspond to data extracted from the literature, exactly from the same source as the authors used for this paper. Zhang et al. also did a polarization map, with no definitive polarimetric structures resolved.

AUTHORS There are differences in our and Zhang's conclusions, and in our opinion they result, most likely, from two issues: (1) a difference in the accuracy of FORS and ACS/WFC polarimeters; (2) selection of the polarimetric data of the other comets.

Regarding the first point, we can first note that the uncertainty we declare is up to 1 dex lower than that of the measurements by Zhang et al.. We are confident that the estimate of the uncertainty of our measurement is correct. We have employed a technique (the beam-swapping technique) which is self-calibrating, and we have performed a number of quality check controls. For instance, the Stokes parameter U/I measured using as a reference direction the perpendicular to the scattering plane is consistent with zero within the declared uncertainty, as expected for symmetry reasons. To further support the validity of our measurements we have now briefly discussed (in the Appendix with supporting material) the null parameters, and discussed that in all but one epoch they are also consistent with zero. FORS2 is a Cassegrain mounted instrument (hence not affected by polarisation introduced by oblique reflections) and is well characterised. We have used it very often, and routinely performed quality checks and characterisation, both in imaging polarimetric mode and in spectro-polarimetric mode. Fossati et al. (2007) had found that in imaging polarimetric mode the instrumental polarisation was $3 \cdot 10^{-4}$ at most.

We believe that the ACS/WFC observations did not allow to confidently conclude if the polarimetric behaviour of comet 2I/Borisov is similar or different from that of Hale-Bopp, and from most of the other comets because of their comparatively larger uncertainties (up to ten times larger than FORS2 uncertainties).

Regarding the second issue, we must emphasise that Kiselev's database, which was used both in our paper and in the poster by Zhang et al., is not homogeneous, as it collects polarimetric measurements obtained with very different filters, of different regions of the various comets, and using different apertures. To produce Fig. 1 we have therefore performed a selection of the data to plot, in particular, as explained in the caption to the Figure, we have used only the data obtained with filters close to the filters employed in our study. We have noticed that the database often includes several measurements of an individual comet obtained at the same phase-angle and with the same filter, but estimated using different apertures. We have now made Fig. 1 using only the points with the smallest aperture value.

In our paper we have added a reference to the Zhang et al. poster with a summary of these considerations at the end of Section 2.1 "Aperture Polarimetry". Additional comments on the Kiselev database have been added in the same Section.

In the Discussion section, the authors comment on the NIR blue spectrum of Borisov, "which was interpreted as evidence for the presence of small particles". I suggest the authors to soften a bit this statement. The fact that the comet presented a negative, blue reflectance spectrum in the NIR when observed close to perihelion, and I cite Lee et al. (2020) "can be caused by an increased abundance of pure water ice or hydrated minerals, which is likely as the cometary activity is transitioning from CO-dominant volatile to H₂O-dominant volatile, and the comet is close to the perihelion". Actually, Lee et al. only rise the possibility of a decrease of dust size in their abstract, where they say "Possible causes of the decreasing slope are an increase in water ice and/or decrease in dust size".

AUTHORS This is correct. We have changed our text to meet this criticism, and we report now the sentence by Lee et al. *verbatim* (third last paragraph of Sect. 4 "Discussion")

A very recent paper from Kim et al. (2020) presents results on images of 2I/Borisov obtained by the HST between October 2019 and January 2020. The authors modeled the dust, finding that, while the coma particle size from October 2019 observations was around 100 μm (consistent with earlier reports), they were not able to model the direction of the dust coma in the November and December observations using the same size, indicating that the coma particle size could not be characterized by a single value. This suggests that particles are ejected anisotropically. They also observed an asymmetry in the coma in the January 2020 observations, that could only be reproduced by assuming particles of 1 mm. Millimeter sized particles were independently identified in a model by Cremonese et al. (2020). I suggest the authors to incorporate these results into their discussion.

AUTHORS Kim et al. (2020) base their conclusion regarding the size of particles on characteristics of the radiation pressure (parameter β) that affects the dust particles and defines their trajectories in the coma. Specifically they say " β is a function of particle size, approximately given by $\beta \sim 1/a$, where a is the particle radius expressed in microns." They do not provide any reference to support this statement. However, the classical paper on radiation pressure by Burns et al. (1979, Icarus 40, 1-48) or more recent papers by Kimura et al. (2002, Icarus 157, 349-361), shows that the dependence of β on particle size is dual: for very small particles β increases with particle size, then starting approximately at particle radii 0.1 – 0.2 μm , it decreases with the rate that may be approximated as $1/a$. Thus, the values of β obtained by Kim et al. (2020) may be related to very big particles (as they assume) as well as very small particles of radius $< 0.1 \mu\text{m}$ in the case of solid particles (Burns et al. 1979) or $< 0.2 \mu\text{m}$ in the case of aggregated particles (Kimura et al. 2002). This makes the values of β estimated by Kim et al. consistent with an assumption of small particles and, thus, with our results. It is possible that other results in Kim et al. paper (e.g., necessity of two populations of particles or mm size particles to explain asymmetry) are also affected by their restricted approach to the selection of β , specifically, by ignoring dependence of beta on the particle structure. We have summarised these considerations in the new version of our paper (again in the third last paragraph of Sect. 3 "Discussion").

Are there other similarities in terms of size, colors, etc., between comet 2I/Borisov and Hale-Bopp? I would like the authors to expand a bit their comparison of these two objects.

AUTHORS In Section "Photometry" we have added a comparison between the photometric colours of the two comets. A comparison between their size is mentioned in the Discussion to emphasise that the polarimetric properties of the two comets must depend on the microscopic structure of the dust aggregates.

In addition, I do not entirely understand the reason to present a comparison with Bennu. Is the polarization curve of Bennu different from that of other low-albedo asteroids? The authors only say that it shows an "unusually steep phase-polarization curve". They should expand this argument, and if so, if Bennu behaves different from other asteroids, even from other "active asteroids", then its observed activity can be used as a potential explanation of this behavior. Please also consider that the level of activity observed in Bennu is very far away for that observed for comets.

AUTHORS We agree that a discussion about the polarimetric properties of asteroid Bennu is outside of the scope of this paper. Therefore the comparison with Bennu in the text and its datapoints in Fig. 1 were removed.

REFeree 2

The manuscript reports the degree of linear polarization and polarimetric color for the interstellar comet 2I/Borisov observed with the ESO/VLT FORS2 instrument. The observations suggest to place the object in a unique category of comets, with closest apparent similarity to Comet C/1995 O1 (Hale Bopp). The uniqueness is explained by the pristine material of the comet, providing us with information about the presumably distant place of origin for the comet.

There are a small number of data points collected at the solar phase angles of 18-28 degrees. Additional observations at smaller phase angles in spring 2020 were lost due to the restrictions set by the Covid-19 pandemic. The data points collected allow the retrieval of the polarimetric slope at the inversion angle, but the scarce data make it impossible to predict the polarimetric behavior at smaller and larger phase angles with any of the existing empirical models. The manuscript can become publishable after revision according to the comments below.

The authors' claim that the polarimetric data are unusual, as stated already in the title, should be quantified by a statistical test. This must be done due to the small number of observed data points (Table 1, Fig. 1).

AUTHORS We have performed a statistical test, and described it in Sect. 4.3, "Statistical tests". In short, we calculate statistical prediction bands for observations around the best-fit phase-polarisation behaviour (Eq 1) for 1P/Halley and the other comets. We use these to show that there is < 1% chance that the 2I/Borisov measurements are consistent with observations drawn from that curve. The findings are summarised in the main text (4th paragraph of Sect. 3: "Discussion").

There is no consensus in the planetary science community about the hypothesis that there would be two classes of comets, that is, low-polarization and high-polarization comets. The authors should offer a more balanced view about the models for cometary polarization.

AUTHORS Our wording in the second paragraph of Section 3 "Discussion" was: "Several studies have divided comets into two polarimetric classes: low- and high-polarisation comets. This different behaviour is ascribed to a very different dust-to-gas ratio in the coma." We have changed the text to say that "This different behaviour is *usually* ascribed..." and mentioned alternative scenarios.

No physical modeling is carried out for the observations in the manuscript. Thus, reviews of existing models are called for. It turns out that some cometary polarization models are reviewed but some are not.

AUTHORS Indeed, the limited range of the phase angles covered by our observations and absence of any features or trends in the coma did not provide sufficient information for any modeling. The Hale-Bopp data were obtained for a larger range of phase angles and a very broad range of wavelengths, including near and thermal infrared, that provided a much better opportunity for interpretation and we refer to these conclusions in the paper. So, to avoid speculative conclusions we decided not to perform any modeling and base our conclusions on the basic physical ideas.

For example, it would be necessary to widen the selection by reviewing and analyzing the following modeling works:

1) Zubko et al. (2015), Characteristics of dust in Comet C/1995 O1 (Hale-Bopp) inferred with polarimetry, The Sixth Moscow Solar System Symposium, Moscow, Russia, pp. 68-70, 2015, openly available, and references in the paper.

AUTHORS We think that the conclusions of this paper are a bit controversial, as the carbonaceous composition of the jets and halo claimed in this paper is not consistent with the bluer color of those features, and contradicts the thermal IR data, which require the Hale-Bopp dust to be of olivine/pyroxene composition and dominated by small particles (Hanner, M.S., et al. , 1997. Thermal emission from the dust coma of comet Hale-Bopp and the composition of the silicate grains. Earth, Moon, and Planets, 79(1-3), pp.247-264.) Therefore, prefer to refer to another Zubko's paper (that is also easier to access): Zubko, E., Videen, G., Hines, D.C. and Shkuratov, Y., 2016. The positive-polarization of cometary comae. Planetary and Space Science, 123, pp.63-76.

2) Markkanen et al. (2018), *Interpretation of the phase functions measured by the OSIRIS instrument for Comet 67P/Churyumov-Gerasimenko. Astrophys. J. Lett. 868, L16.*

AUTHORS We have included this reference in our paper.

The former assesses Hale-Bopp that is clearly important for the present study and the latter assesses the Rosetta mission target comet, for which very large coma particles were measured by Rosetta in situ."

AUTHORS Some additional references to the modelling efforts have been requested also by Referee 3 and have been included in the Introduction. We would like to comment, however, that probably there is no much similarity between the dust properties of 2I/Borisov and those of the dust of 67P. The large particles found by the Rosetta mission are typical for old comets like all Jupiter family comets are. The large particles are a result of a long processing of the cometary surface by solar radiation and charged particles. For new comets or interstellar objects small particles should be more realistic (for a more detailed comparison of the dust in young and old comets see, e.g., Lisse, C. M., A'Hearn, M. F., Fernández, Y. R., & Peschke, S. B. 2002, in *Dust in the Solar System and Other Planetary Systems*, ed. S. F. Green, I. P. Williams, J. A. M. McDonnell, & N. McBride (Oxford: Pergamon), 259. and in Kolokolova et al. (*Astrophys. J.*, 463, 2007). Although this discussion is outside of the scope of the paper we have cited the work by Markkanen et al. (2018) as a further example of modelling (third last paragraph of the Introduction).

Additionally, there are two minor points to be accounted for:

1) *Currently, both "solar system" and "Solar System" appear frequently in the text.*

AUTHORS We have adopted Solar System throughout the text.

2) *"Trans-Neptunian" should rather read "transneptunian" (cf. "extraterrestrial").*

AUTHORS Fixed.

REFeree 3

This paper, submitted to Nature Communication, presents unique results about polarimetric measurements on interstellar comet 2I/Borisov and their interpretation. It is well structured and pleasantly written by experts on polarimetric observations of surfaces of small bodies and of dust particles ejected by cometary nuclei, and certainly deserves publication in Nature Communications. While the observations and their interpretation are remarkable, it would be better to:

1. *Provide some background on polarimetric observations and their significance, for readers not so much familiar with such topics.*
2. *Tentatively find more information through Figure 3.*
3. *Extend the discussion, in terms of possible contamination of the polarization (induced by solar light scattering on dust particles) by gaseous emissions. Also, update the discussion, with complementary observations of a new comet presenting, as C/1995 O1 Hale Bopp, a very high polarization, and possibly with references to a couple of recent papers about cometary dust properties.*

AUTHORS To meet the request of point 1 we have updated the Introduction with some general considerations about the use of polarimetry in solar system science (second paragraph). The remaining requests of the referee are further explained/expanded below, so our answer will follow the next points.

1. Introduction (2nd paragraph)

1.1. *While comets and asteroids are mentioned, it could be better to explain that the information provided by polarization induced by scattering on surfaces (asteroids or cometary nuclei) is easier to interpret than the polarization induced by the scattering on huge dust clouds, such as cometary comae or tails. In the latter case, the signal corresponds to different conditions along the line-of-sight, such as the distance to the nucleus; the dust particles may exhibit quite different properties in dust jets and possibly in innermost comae (e.g., Hadamcik & Levasseur-Regourd, Icarus, 2003; Kiselev et al. review, In Polarimetry of stars and planetary systems, Kolokolova et al. Eds, CUP, 2015).*

AUTHORS We are reluctant to state that modelling the polarisation produced by scattering from a surface is *easier* than the modelling of the polarisation induced by scattering from dust aggregates, as both tasks have their own challenges. However it is certainly appropriate to highlight these difficulties in the interpretation of comet polarisation, and we have added some relevant comments in the text (third paragraph of the Introduction).

1.2. *It could also be explained that polarimetric observations, complemented by numerical and experimental simulations, may provide information on the size and size distribution, on the morphology and porosity, and on the complex refractive indices, whence clues to the composition (again in Polarimetry of stars and planetary systems, 2015, reviews by, e.g., Mishchenko; Levasseur-Regourd et al.; Cellino et al.; Kiselev et al.).*

AUTHORS We have added this explanation in the text (third paragraph of the Introduction).

2. Polarimetry along the tails

- 2.1. *Nice Figure 3 would be improved by indications of the orientations of the dust and gas tails.*
- 2.2. *It would be of interest to provide a B/W figure with iso-contours of the coma (possibly removing the right part of Fig. 3), which could enhance some asymmetries.*
- 2.3. *The orientation of the dust trail (along the trajectory) could be searched, in the (unlikely) case of some heterogeneities suggesting the presence of large fragments.*

AUTHORS Figure 3 has been modified to incorporate these referee’s suggestions, but we could not find any evidence of dust trail for this object.

3. Observations and their discussion

3.1. The observations are obtained for values of the phase angle between 18° and 29° , that is to say in the so-called inversion region. Results on the inversion angle at given wavelengths could be compared with those obtained on asteroids and other media.

AUTHORS In Section 2.1. "Aperture Polarimetry", when we present our new measurements, we have included a short discussion about the fact that different objects of the Solar System exhibit different values of the inversion angle.

3.2. The errors bars of the values obtained through the (quite large) R and I filters should be discussed, taking into account the possible contamination by gaseous emissions, which can only be avoided through narrow interference filters dedicated to cometary dust observations. Indeed, most cometary dust polarimetric observations are done with narrow interference filters, as typically provided by ESA for Churyumov-Gerasimenko observations (see, e.g. Hadamcik et al., MNRAS 2017).

AUTHORS We had already commented that in general it would be useful to adopt filter specifically designed to avoid emission lines, (which are not available at FORS2) but 2I/Borisov does not show prominent emission lines that may contaminate the R and I filters, and we have further highlighted this at the beginning of Sect. 2 "Observations".

3.3. The bandwidths of all filters mentioned in Table 1 should be listed in the Table or its caption.

AUTHORS The bandwidth and effective wavelength are so important that we have specified them in the text. We have added the same information in the Caption to Table 1.

4. Comparison of polarimetric properties of 2I/Borisov with those of other comets An significant result is missing in this part of the discussion, since comet C/1995 O1 Hale-Bopp is not the single one to have shown a very high polarization. At least an other comet, C/1999 S4 LINEAR, has been found to present a high polarization (above 100° phase angle). It can be noticed on Fig. 4 of Hadamcik & Lvasseur-Regourd, Icarus 2003, i.e. reference # 30 in the paper presently reviewed.

AUTHORS Data of comet C/1999 Linear were obtained at a very different phase-angle range than 2I/Borisov and even Hale-Bopp. It is not easy to compare these datasets. Furthermore, comet C/1999 Linear had a normal polarization until it started breaking apart, and all the high values relate to the observations during a very wild comet disruption, when probably there were many small dust particles released during this event. We added some considerations about this in Sect. 3 "Discussion".

5. Minor comments

Grammar and spelling possibly needing to be checked. E.g., "Measurements of cometary linear polarisation provideS information about..."

AUTHORS We read the manuscript carefully and we hope to have eliminated the various typos.

Figure 1. While the most visible data points and fits correspond to Bennu and Hale-Bopp (in dark blue), the new and fascinating data points related to 2I/Borisov should be more conspicuous.

AUTHORS If we expand the symbols used for 2I/Borisov, then errorbars will become invisible (i.e., smaller than the symbols), so we have reduced the size of the symbols used for the other comets.

Figure 2. The colour code is completely different from the one used in Fig. 1. It may be delicate for the reader, since those two figures could be close to one another in the paper.

AUTHORS We have changed the colour coding to make it consistent with Fig. 1.

It would be better to follow the terminology now accepted for cometary dust particles (Güttler et al., A&A 630 2019). The wording “grains” should be used for the irregular and compact monomers, i.e. tiny building blocks, of the dust particles. The dust particles are irregular and fractal “aggregates” or “agglomerates”, most likely with a wide range of porosities (e.g. Mannel et al., also within A&A 630, 2019). The wording “particulates” could be avoided.

AUTHORS Fixed.

REVIEWERS' COMMENTS

Reviewer #1 (Remarks to the Author):

The authors have successfully addressed all the suggested comments and critics and have adapted the main text accordingly. Therefore I fully recommend the revised version of the manuscript for publication.

Reviewer #2 (Remarks to the Author):

REVIEW OF MANUSCRIPT:

S. Bagnulo et al., Unusual polarimetric properties for interstellar comet 2I/Borisov, submitted to Nature Communications, revised manuscript NCOMMS-20-43192A

Thanks to the revision by the authors, the manuscript has improved substantially. I can recommend publication after minor but potentially somewhat elaborous revision.

The revision request concerns the confusing filter names v_{HIGH} , R_{SPECIAL} , I_{BESS} , etc. First, this way of phrasing subscripts is not that elegant. Second, it is confusing to rename the filters to V , R , and I in the italic font, because of the Stokes parameters I , Q , U , V in the same font. Third, the roman font is excluded due to it being reserved, naturally, for the Johnson-Cousins UBVRI system, as the

authors write in Sect. 4.2.

Karri Muinonen Helsinki,

January 18, 2021

Reviewer #3 (Remarks to the Author):

The efforts of the authors in revising the manuscript are much appreciated, all the more because suggestions and requests were coming from three different referees.

Point 1

Appropriate, with more background tentatively provided within the introduction.

Point 2

- The new version of Figure 3, including isophotes is presented.
- It is nevertheless necessary to “review the second part of the caption”. The last sentence is indeed quite difficult to understand, with the exact meaning of the red dotted line not mentioned, and the main orientations of the dust and gas tails not clearly visible.

Point 3

- The comment about the inversion angle is appreciated.
- It should nevertheless be noted that “the relationship between the slope of the polarimetric curve at the inversion phase angle and the albedo is established for asteroidal surfaces, and not for active comets”, the observed polarization of which is likely to mostly come from the dust coma (and possibly tail).
- The addition of the characteristics of the filters also in the caption of Table 1 is appreciated.

Minor comments

- Perfect
- Nevertheless, there is still once “coma dust grains” that should be coma dust particles, within Introduction (2nd paragraph, 1st line).

REFeree 1

The authors have successfully addressed all the suggested comments and critics and have adapted the main text accordingly. Therefore I fully recommend the revised version of the manuscript for publication.

AUTHORS We thank the referee for looking once again to our paper and for their positive comments.

REFEREE 2

Thanks to the revision by the authors, the manuscript has improved substantially. I can recommend publication after minor but potentially somewhat laborous revision. The revision request concerns the confusing filter names v_HIGH , $R_SPECIAL$, L_BESS , etc. First, this way of phrasing subscripts is not that elegant. Second, it is confusing to rename the filters to V , R , and I in the italic font, because of the Stokes parameters I , Q , U , V in the same font. Third, the roman font is excluded due to it being reserved, naturally, for the Johnson-Cousins $UBVRI$ system, as the authors write in Sect. 4.2.

AUTHORS: We thank the referee for looking once again to our paper and for his positive comments. We agree that the filter notation is unfortunate, but it is the one used in the FORS2 webpages and user manual. We do not have a good solution on how to refer to them without using their exact technical definition, at least when mentioned for the first time. We understand that VRI may be potentially confusing with Stokes $IQUV$, and we propose to refer to FORS2 filters with the symbols V_F , R_F, I_F after their first introduction with their technical names.

REFEREE 3

The efforts of the authors in revising the manuscript are much appreciated, all the more because suggestions and requests were coming from three different referees.

Point 1: Appropriate, with more background tentatively provided within the introduction.

Point 2: The new version of Figure 3, including isophotes is presented. It is nevertheless necessary to "review the second part of the caption". The last sentence is indeed quite difficult to understand, with the exact meaning of the red dotted line not mentioned, and the main orientations of the dust and gas tails not clearly visible.

AUTHORS: We thank the referee for looking once again to our paper and for her positive comments. Because of the Wollaston strips, our images do not cover a large field of view, and cannot probe the entire comet ejecta. Nevertheless, the false colour image of the top panel shows clearly that the gas tail is oriented along the anti-solar direction, then the dust tail goes along the horizontal direction. The red dotted line in the bottom line shows this behaviour in more detail (and better than an arrow that would make the figure a bit too crowded with labels). We have updated the caption as follows: "The red dotted line traces the comet tail, which at shorter distance from the photocentre is directed along the anti-solar direction, and at larger distances lies in between the anti-solar and anti-velocity direction." We hope that this represents a better explanation. The Figure has also been modified to better show the spatial scale.

Point 3: The comment about the inversion angle is appreciated. It should nevertheless be noted that "the relationship between the slope of the polarimetric curve at the inversion phase angle and the albedo is established for asteroidal surfaces, and not for active comets", the observed polarization of which is likely to mostly come from the dust coma (and possibly tail). The addition of the characteristics of the filters also in the caption of Table 1 is appreciated.

AUTHORS: Indeed, our paper states: "However, the albedo-polarisation relationship cannot be strictly applied to cometary comae, because of the very different physics of light scattering by rather densely packed layers of particles in the former, and by tenuous clouds of particles in the latter." We have added "and tails" after "comae", and replaced "the latter" with "active comets".

Minor comments. Perfect. Nevertheless, there is still once "coma dust grains" that should be coma dust particles, within Introduction (2nd paragraph, 1st line).

AUTHORS We have replaced "grains" with "particles".